# Insulin Sensitivity Controls Activity of Pathogenic CD4+ T Cells in Rheumatoid Arthritis

**DOI:** 10.3390/cells13242124

**Published:** 2024-12-22

**Authors:** Malin C. Erlandsson, Eric Malmhäll-Bah, Venkataragavan Chandrasekaran, Karin M. E. Andersson, Lisa M. Nilsson, Sofia Töyrä Silfverswärd, Rille Pullerits, Maria I. Bokarewa

**Affiliations:** 1Department of Rheumatology and Inflammation Research, Institute of Medicine, University of Gothenburg, 41346 Gothenburg, Sweden; malin.erlandsson@rheuma.gu.se (M.C.E.); rille.pullerits@rheuma.gu.se (R.P.); 2Rheumatology Clinic, Sahlgrenska University Hospital, 41345 Gothenburg, Sweden; 3Department of Clinical Immunology and Transfusion Medicine, Sahlgrenska University Hospital, 41346 Gothenburg, Sweden

**Keywords:** insulin, CD4+ cells, senescence, arthritis, interferon

## Abstract

Hyperinsulinemia connects obesity, and a poor lipid profile, with type 2 diabetes (T2D). Here, we investigated consequences of insulin exposure for T cell function in the canonical autoimmunity of rheumatoid arthritis (RA). We observed that insulin levels correlated with the glycolytic index of CD4+ cells but suppressed transcription of insulin receptor substrates, which was inversely related to insulin sensitivity. This connection between insulin levels and the glycolytic index was not seen in CD4+ cells of healthy controls. Exposure of CD4+ cells to insulin induced a senescent state recognized by cell cycle arrest and DNA content enrichment measured by flow cytometry. It also resulted in accumulation of DNA damage marker γH2AX. Insulin suppressed IFNγ production and induced the senescence-associated secretome in CD4+ cell cultures and in patients with hyperinsulinemia. Inhibition of JAK-STAT signaling (JAKi) improved insulin signaling, which activated the glycolytic index and facilitated senescence in CD4+ cell cultures. Treatment with JAKi was associated with an abundance of naïve and recent thymic emigrant T cells in the circulation of RA patients. Thus, we concluded that insulin exerts immunosuppressive ability by inducing senescence and inhibiting IFNγ production in CD4+ cells. JAKi promotes insulin effects and supports elimination of the pathogenic CD4+ cell in RA patients.

## 1. Introduction

Insulin signaling plays a central role in cell metabolism. Acting through its receptor, insulin promotes glucose uptake by activating glucose transport proteins, stimulates glycogen synthesis via GSK3, and drives protein transcription through pathways such as PI3-kinase, AKT1, MAPK, and mTOR (Figure 1A). The insulin receptor (INSR) is highly expressed in tissues that depend heavily on glucose metabolism, including adipose tissue, hepatocytes, brain, and skeletal muscle.

Systemic inflammation during severe infections, as well as exposure to pro-inflammatory cytokines like TNF and IFNγ, is closely associated with a reduction in insulin’s intracellular effects, a condition referred to as insulin resistance [1,2,3]. The effects of insulin resistance differ significantly between tissues. In adipose tissue, insulin resistance disrupts lipolysis, causing increased cell size and altered protein release profiles. In skeletal muscle, it leads to the accumulation of ceramides and diacylglycerols, contributing to mitochondrial dysfunction, oxidative stress, and reduced muscle strength. In the liver, insulin resistance impairs gluconeogenesis, reducing energy supply and promoting cellular senescence in hepatocytes, adipocytes, and macrophages [4,5]. This state of aging-related dormancy is characterized by telomere shortening, loss of proliferative capacity, and resistance to apoptosis, often culminating in liver steatosis. In the brain, where efficient glucose metabolism is critical, insulin resistance disrupts neuronal function, synaptic plasticity, and energy balance, contributing to neurodegeneration. Signaling through the IR/IGF1R heterodimer is identified as a mitogenic driver in tumor cells, linking it to the development of several cancer types. Consequently, inducing insulin resistance is sometimes employed as a therapeutic strategy in cancer treatment [6].

Insulin resistance is often accompanied by excessive insulin production from pancreatic beta cells. Clinically, hyperinsulinemia serves as an early indicator of metabolic dysfunction, poor glucose and lipid profiles, and an increased risk of developing obesity, metabolic syndrome, and type 2 diabetes (T2D) [7,8]. While chronic hyperinsulinemia is typically associated with negative outcomes, it has therapeutic applications in specific cases, such as managing postprandial glucose spikes in type 1 diabetes (T1D) or redistributing elevated potassium levels to prevent cardiac arrhythmias.

In autoimmune diseases like rheumatoid arthritis (RA), chronic inflammation is linked to increased insulin production. Interestingly, the prevalence of hyperinsulinemia in RA is significantly higher than in the general population or even among patients with systemic lupus erythematosus [9,10,11], though its role in the disease remains unclear.

In T cells, which play a central role in RA pathogenesis, insulin activates signaling through the insulin and IGF1 receptor heterodimer. This promotes glucose uptake, which is critical for energy supply in effector immune functions and differentiation. Although IGF1R expression is dominant, the insulin receptor plays a role in maintaining T cell effector function [12,13]. Experimental insulin resistance, either by deleting the insulin receptor [12,13] or by inhibiting IRS adaptor proteins [13,14], reduces antigen-specific T cell responses, including proliferation. It also shifts the balance between pro-inflammatory and regulatory T cell subtypes, alleviating experimental colitis and autoimmune arthritis.

While substantial evidence supports insulin’s pro-inflammatory effects in immune cells, some studies suggest it can also dampen inflammation. Insulin administration in T2D has been shown to reduce C-reactive protein levels and the oxidative capacity of neutrophils [15]. Activation of the PI3-kinase-AKT pathway inhibits FOXO transcription factors, which are involved in cell proliferation and energy metabolism, while stimulating the anti-inflammatory factor NRF2 through ERK signaling. Excessive INSR signaling can also suppress Treg function, improving glucose tolerance and insulin sensitivity in conditions such as obesity and aging [16].

This study aimed to prospectively investigate the impact of insulin signaling and hyperinsulinemia on the function of CD4+ T cells in RA. We analyzed the transcriptional footprint of insulin signaling in CD4+ cells and experimentally assessed insulin’s effects on cell cycle progression and the senescence profile of CD4+ cells. Finally, we examined how JAK/STAT inhibition affects insulin signaling, both under experimental conditions and in RA patients treated with JAK inhibitors.

## 2. Patients and Methods

### 2.1. Study Material

This study used two independent patient cohorts (Appendix A), which were collected at the Rheumatology Clinic of Sahlgrenska University Hospital, Gothenburg, Sweden. Cohort one was collected between 10 January 2022, and 14 March 2023. It consisted of 16 non-diabetic untreated RA patients and 69 non-diabetic subjects having no rheumatic diseases (healthy controls, HC) randomly selected among the 1st-visit patients with musculoskeletal complains. Cohort two was collected between 7 October 2019 and 26 October 2020. It consisted of 56 RA patients with established disease. All RA patients fulfilled the 2010 EULAR/ACR classification criteria [17]. The disease activity score (DAS28) of RA was calculated based on assessment of 28 tender and swollen joints and erythrocyte sedimentation rate.

Among the RA patients within cohort 2, 24 were treated with the Janus kinase inhibitors (JAKi) (Appendix A). Twelve patients combined JAKi with methotrexate, four with other antirheumatic drugs, and one each with abatacept, tocilizumab, or sarilumab. Among the non-JAKi-treated patients, fourteen were treated with methotrexate monotherapy, ten with TNF inhibitors, and one with tocilizumab. Two of the JAKi-treated and three of the non-JAKi-treated patients used oral corticosteroids. Sixteen patients had neither methotrexate, JAKi, nor oral corticosteroids. None of the control subjects were using immune suppressive drugs or oral corticosteroids.

For the in vitro experiments, cells from additional healthy non-diabetic subjects (26 female, 14 male, age 46 ± 13 years) were used. All study persons gave their written informed consent to physical examination and blood sampling.

This study was approved by the Swedish Ethical Review Authority and was registered at ClinicalTrials.gov with ID NCT03449589, NCT03444623.

### 2.2. Serological Measurements

Blood sampling for serum and EDTA plasma preparation was performed between 7 am and 10 am from the cubital vein. Samples were aliquoted and stored at −70 °C.

Serum IGF1 and complete blood counts were measured at the Laboratory of Clinical Chemistry, all within routine patient care. Plasma insulin levels were measured by sandwich ELISAs (DY8056, R&D Systems, Minneapolis, MN, USA). Serum rheumatoid factor was measured by rate nephelometric technology (Beckman Immage 800, Beckman Coulter AB, Brea, CA, USA). Antibodies against anti-cyclic citrullinated protein were measured by the automated multiplex method (Bioplex2200, Biorad, Hercules, CA, USA) at the accredited Laboratory of Clinical Immunology, Sahlgrenska University Hospital.

### 2.3. Cell Isolation and Culturing

Human peripheral blood mononuclear cells were isolated from the peripheral heparinized blood by density gradient centrifugation on Lymphoprep (Axis-Shield PoC As, Oslo, Norway). CD4+ T cells were isolated by positive selection (11331D, Invitrogen, Waltham, MA, USA), and cultured at a cell density of 1.25 × 10^6^ cells/mL in wells coated in anti-CD3 antibody (0.5 microg/mL; OKT3, Sigma-Aldrich, St. Louis, MO, USA) in the Roswell Park Memorial Institute (RPMI1640) medium (Gibco, Waltham, MA, USA) supplemented with 5% fetal bovine serum (Sigma-Aldrich), 4 mM Glutamax (Gibco), 50 mM β2-mercaptoethanol (Gibco), and 50 mg/mL gentamycin (Sanofi-Aventis, Paris, France) at standard conditions of temperature, CO_2_ pressure, and humidity. The culture media was supplemented with insulin (0 and 10 nM, Humalog 100 U/mL, Eli Lilly, Indianapolis, IN, USA) for 48 h at 37 °C and/or tofacitinib (0 or 10 µM, Selleck Chemicals, Houston, TX, USA).

### 2.4. DNA Content and Cell Cycle Analysis

CD4+ T cells were stained with CellTrace violet (Invitrogen Life Technologies, Waltham, MA, USA) according to the manufacturer’s instruction to follow cell proliferation and stimulated as above for 48 h with or without insulin or a non-selective JAKi tofacitinib. At harvest, supernatants were collected, cells were permeabilized for 1 h, 4 °C with cytofix/cytoperm (BD Biosciences, Franklin Lakes, NJ, USA) and incubated overnight, 4 °C, with 20 μg/mL 7-Aminoactinomycin D (7AAD) (Invitrogen) in perm/wash (BD Biosciences). Stained cells were washed 3 times with perm/wash and resuspended in 200 μL FACS buffer. Cells were acquired in FACS Verse (BD Biosciences). Analysis of the acquired data was performed using the Tree Star FlowJo software v.10 using the built-in cell cycle analysis tool, Watson model with constraints, CV(G2) = CV(G1) [18]. The acquired CD4+ cells were gated on 7AAD+ and separated by the forward cell scatter into large-cell size and small-cell size populations (Appendix A).

### 2.5. Immunocytochemistry of DNA Damage

Human monocytic THP1 cells (TIB-202, ATCC, Manassas, VA, USA) were seeded with a density of 5 × 10^5^/mL on glass chamber slides (Thermo Fisher Scientific, Waltham, MA, USA) precoated with poly-L-lysine (Sigma-Aldrich, Saint Louis, MO, USA). Cells were treated with insulin (Humalog, Eli Lilly), 0, 10 or 25 nM overnight, at 37 °C, 5% CO_2_. At harvest, cells were fixed with 4% formalin and permeabilized for 3 h with 3% normal goat serum and 1% TritonX100. Slides were covered with antibodies against H2AX phosphorylated at Ser139 (γH2AX, mouse, 05-636, Merck Millipore, Burlington, MA, USA) and isotype controls (mouse IgG Dako X0931) diluted in blocking buffer and incubated over night at 4 °C. Secondary antibodies donkey-anti-mouse AF488 (Invitrogen A-21202) were applied for 2 h at room temperature. Autofluorescence was blocked with 0.5% Sudan Black B (Sigma-Aldrich) in 70% ethanol for 20 min at room temperature. Nuclei were stained with Hoechst 34,580 (NucBlue Live Cell Stain; Thermo Fisher Scientific) for 20 min and mounted with ProLong Gold antifading mounting reagent (Invitrogen).

### 2.6. Confocal Imaging and Analysis

Fluorescence microscopy was performed using the confocal imaging system Leica SP8 (Leica Microsystems, Wetzlar, Germany) with sequential acquisition using a 40× oil objective and up to 10× digital zoom. The images were acquired at high resolution (1.5 × digital zoom), viewing 15–30 nuclei per image. Within each sample, γH2AX-positive foci were enumerated in 2–3 images, resulting in 44–63 nuclei per treatment. Images were analyzed with ImageJ version 2.9 within 8-bit composite images [19]. The threshold was adjusted to optimize identification of positive spots. Nuclear area was defined by thresholding the Hoechsts (blue) image and exporting the results to ROI. The number of γH2AX-positive foci in each nucleus stained with Hoechsts was estimated after noise reduction by de-speckling using the ImageJ feature Find Maxima.

### 2.7. Cytokine Measurement

Cytokine levels in serum and supernatants were measured by sandwich ELISAs for VEGF (DY293B), IL-8 (DY208), and survivin (DYC647) (R&D Systems, Minneapolis, MN, USA), and IL-6 (M1916), TNF (M1923), and IFNγ (M1933) (Sanquin, Amsterdam, The Netherlands), as previously described [20,21].

Dot blot cytokine array Proteome Profiler™ Array, Human Cytokine Array Panel A (ARY005, R&D Systems), was used for semi-quantitative measurement of 36 cytokines and chemokines in the supernatants of cultured cells, as previously described [22]. In short, the array membranes precoated with capture antibodies were soaked in the supernatant of the insulin-stimulated and control cells pooled in equal volume from each cell culture.

### 2.8. Gene Expression Analysis by Conventional RT-PCR

RNA from CD4+ cell cultures was prepared using the micro mRNA kit (Norgen, Thorold, ON, Canada). Complementary DNA was prepared using the High-Capacity cDNA Reverse Transcription Kit (Applied Biosystems, Foster City, CA, USA). Quantitative PCR was performed with SYBR Green qPCR Mastermix (Qiagen, Hilden, Germany) using a ViiA™ 7 Real-Time PCR System (Applied Biosystems). Expression was normalized to the reference gene ACTB (TATAA Biocenter, Gothenburg, Sweden). Melting curves for each PCR were performed between 60 °C and 95 °C to ensure specificity of the amplified product. The results are expressed in relative quantity (RQ) to the expression level in the control cells using the ddCt-method.

### 2.9. Primer Design

Primers were designed in-house using the Primer3 web client (https://primer3.ut.ee/, accessed on 17 April 2022). Amplicon and primer size was limited to 60–150 and 18–24 base pairs, respectively. Melting temperature was set between 60 and 63 °C, max poly-X to 3, and GC-content was limited to 40–60%. Primers were checked in NetPrimer (https://www.premierbiosoft.com/netprimer/, accessed on 17 April 2022) for possible hairpin and primer–dimer structures. Correct binding of primers was validated in UCSC In-Silico PCR (http://genome.ucsc.edu/cgi-bin/hgPcr, accessed on 17 April 2022) against the GRCh38/hg38 human genome assembly. Primer sequences are available in Appendix A.

### 2.10. Transcriptional Sequencing (RNA-Seq)

RNA from CD4 cell cultures was quality controlled by Bioanalyzer RNA6000 Pico on Agilent 2100 (Agilent, St. Clara, CA, USA). Deep sequencing was done by RNA-seq (Hiseq2000, Illumina, San Diego, CA, USA) at the LifeScience Laboratory, Huddinge, Sweden. Raw sequence data were obtained in Bcl-files and converted into fastq text format using the bcl2fastq program from Illumina. Fastq-files and the processed reads are deposited in Gene Expression Omnibus (GEO) at the National Centre for Biotechnology Information with the accession codes GSE201669 and GSE282517.

### 2.11. Transcriptional Sequencing Analysis

Mapping of transcripts was conducted using Genome UCSC annotation for the hg38 human genome assembly. The differentially expressed genes (DEGs) were identified by R-studio using the Bioconductor package, “DESeq2” version 1.26.0 [23].

### 2.12. Bioinformatic Analysis

Functional enrichment analysis for the Gene Ontology defined biological processes (GO:BPs) and molecular function (GO:MF) was done for the top 500 protein coding DEGs (expression basemean > 5, adj *p*-value < 0.05) in both directions using gene set enrichment analysis (GSEA, Broad Institute, Cambridge, MA, USA). GO:BPs were limited to those containing <1500 genes from the analyzed category and utilized the false discovery rate (FDR) 5% *p*-value adjustment. TF targets were identified as genes containing one or more binding sites for the given TF in their promoter regions (TSS –1000, +100 bp) as identified by GTRD version 20.06 ChIP-seq harmonization.

### 2.13. Statistics

The glycolytic index (GI) was calculated as the sum of the normalized expression (DESeq2) of genes representing the glycolytic pathway [24]. Differential expression analysis was performed by DESeq2 [23] in samples split by median GI and by plasma insulin levels of 157 pmol/L and comparing RA patients treated with JAKi to patients with other treatments (Appendix A). To prepare the log2 fold change, heatmaps with nominal *p*-values were prepared using the ComplexHeatmap package [25]. Correlation heatmaps were prepared using the R package Corrplot [26].

Statistical analysis was conducted in R and in Graphpad Prism (version 10.4, Boston, MA, USA). Paired comparison between groups was conducted by the Wilcoxon test, and unpaired comparison by the Mann–Whitney U test for 2 groups and the Kruskal–Wallis test when more than 2 groups were compared. Correlations were conducted with Spearman’s non-parametric correlation. The *p*-values below 0.05 were considered significant.

## 3. Results

### 3.1. Distinct Insulin Sensitivity of CD4+ Cells from RA Patients and Healthy Controls

To investigate the insulin signaling in CD4+ cells, we compared transcriptomes of these cells in 16 treatment naïve RA patients and 69 healthy controls (HC) matched with RA patients by age and gender (Appendix A).

We found that a summarized transcription of glycolytic enzymes G6PD, HK3, PFKFB3, PFKFB2, ALDOA, PGM1, LDHA, PGAM1, ENO1, and GAPDH comprising the glycolytic index (GI) was significantly lower in RA patients compared to HC (Figure 1B). Additionally, plasma insulin levels were strongly correlated to the glycolytic index (GI) of CD4+ cells of RA patients, which was not seen in HC (Figure 1C). This difference in the insulin correlation strength was found despite the comparable plasma insulin levels and was supported by the increasing proportion of INSR in CD4+ cells, measured by the INSR/IGF1R expression ratio, of both RA patients and HC (Figure 1D,E; Spearman rho 0.447 RA and 0.549 HC, respectively). Notably, both the insulin levels and GI presented an inverse correlation with IRS proteins, which act as a check-point of cellular insulin sensitivity, and showed a lineal repression of IRS proteins in CD4+ cells where GI was high (Figure 1E). The insulin levels and GI were positively correlated with the markers of systemic inflammation, including CRP, WBC, and platelet counts, and serum levels of IL6, IFNγ, and IL8. This association was stronger in HC and weaker in RA patients. In contrast to insulin, the IGF1 levels were inversely correlated to GI and INSR/IGF1R ratio, and, also, to the markers of systemic inflammation (Figure 1E).

### 3.2. Glycolytic Index Dependent Diversity in Intracellular Processes of CD4+ T Cells

Insulin signaling promotes glucose transport, cell survival, transcriptional regulation, and glycogen synthesis. In CD4+ cells, these effects of insulin were associated with GI (Figure 1F) and accounted for the upregulation of glucose transporter SLC2A1, pyruvate dehydrogenase kinase PDK1 leading to AKT serine/threonine kinase AKT1, and suppressors of cytokine signaling SOCS1-3 leading to activation of p38 MAP-kinases coded by MAPK11 and MAPK12. The combined activation of p38 and the AP-1 transcription complex consisting of JUN and FOS controlled DNA metabolism [27,28] and restrained the FOXO and TCF7/LEF1 genes, the major target of insulin in gene transcription control (Figure 1F). In line with this, PI3-kinase signaling was negatively affected by the inhibition of IRS1 and IRS2 genes, which was translated in a repression of PIK3CA and PIK3CB coding for p110 catalytic units, and PIK3R1 coding for the regulatory subunit of the PI3-kinase complex (Figure 1F). The effect of insulin signaling was diverted in CD4+ cells of RA patients and HC, which was consistent with the significantly lower GI in RA patients (Figure 1B). In RA, we found a significantly lower SLC2A1 and PDK1, leading to a relative deficiency in AKT1, and lower transcription of the SOCS and p38 genes. Together with this, RA CD4+ cells presented higher expression of transcriptional regulators FOXO1, FOXO4 and TCF7, and LEF1 (Figure 1F). These observations suggested that CD4+ cells of RA patients had the transcriptional pattern of an intrinsic insulin resistance, while the HC CD4+ cells appeared to be more sensitive to insulin. Analysis of the genes highly expressed in association with GI identified their dependence from the CCAT Enhancer binding protein zeta (CEBPZ), which is important in cell cycle and proliferation response; the death inducer obliterator 1 (DIDO1), which maintains cellular self-renewal; histone modifiers SETD1A and KAT1A, which orchestrate mitosis and promote DNA damage responses; and the trans-activator of the MHC gene CIITA required for the proper T cell differentiation (Figure 1G). Consequently, this linked them to the biological processes of Cell Cycle (GO:0007049, FDR 1.82 × 10^−42^), Chromosomal organization (GO:0051984, FDR 2.17 × 10^−52^), and DNA repair (GO:0006281, FDR 8.34 × 10^−33^) (Figure 1H). The genes upregulated in association with low GI were controlled by transcription factors ZNF740 and RAG1, which are important for the establishment of adaptive immune responses (Figure 1G). They were linked to the biological processes of RNA polymerase II dependent transcription (GO:0006369, FDR 2.63 × 10^−44^), Positive regulation of RNA metabolism (GO:0051254, FDR 6.16 × 10^−20^), and DNA damage response (GO:0006974, FDR 9.04 × 10^−13^) (Figure 1H).

Indeed, the GI of CD4+ cells showed a strong positive correlation to cell cycle control and DNA damage sensing CDK1/2, BRCA1/2, and chromosomal passenger complex proteins AURKB, INCENP, CDCA8, and BIRC5 (Appendix A). Remarkably, the genes regulating cell cycle and those mediating a balance between DNA damage response and DNA repair appeared enriched in association with both high and low GI of CD4+ cells, which illustrated the importance of sufficient insulin signaling for these critical survival mechanisms in CD4+ cells.

### 3.3. Insulin Induces Pro-Senescent Effects in CD4+ Cells

Based on the observed association between the insulin signaling and cell cycle control, and recently reported senescence-inducing properties of insulin in hepatocytes [18], we investigated functional effects of insulin on anti-CD3-activated CD4+ cells of healthy individuals, through flow cytometry.

Comparing CD4+ cells within the large-size cell (LSC) and small-size cell (SSC) subsets (Appendix A), we found that SSC had a significantly lower DNA content reflected by 7AAD dye. In response to insulin, the SSC subset significantly increased the DNA content (Figure 2A), which occurred in the G1 phase and obstructed the cell cycle progress to the S phase (Figure 2B,C), which suggested that insulin induced a senescent state in SSCs. This DNA content enrichment was not observed in the LSCs (Figure 2A).

Insulin stimulation had no significant effect on the proliferation rate of the cultured CD4+ cells visualized by dilution of the CellTrace Violet dye content (Figure 2D). To confirm development of the senescent state, we exposed THP1 cells to increasing doses of insulin (0, 10 and 25 nM) and measured formation of γH2AX foci, encounter of DNA damage. Confocal imaging of the insulin-exposed cells revealed a dose-dependent increase in the proportion of γH2AX-stained areas in the nuclei, witnessing accumulation of unresolved DNA damage (Figure 2E,F).

Investigating the insulin effect on transcription and cytokine production by CD4+ cells, we found that insulin suppressed IRS1, PIK3CG, STAT5A, and ABL1 mRNA levels (Figure 2G), promoting a decrease in insulin sensitivity. Screening cytokine production in supernatants of CD4+ cell cultures by a cytokine dot array, we found that the insulin stimulation inhibited production of IFNγ, IL8, CCL3/4, RANTES, and GM-CSF by CD4+ cells (Figure 2H). Consistent with the results of the dot array and low mRNA levels of IFNγ (Figure 2G), enzyme-linked assay demonstrated that insulin significantly suppressed IFNγ and TNF production in CD4+ cells, while increasing senescence-associated cytokine IL6 (Figure 2I).

Together, these functional results demonstrated a pro-senescent effect of insulin on CD4+ cells characterized by cell cycle stagnation in the G1 phase, enrichment of DNA damage, and increasing IL6 production.

### 3.4. Chronic Hyperinsulinemia Is Associated with Inhibition of the Adaptive Immune Response in CD4+ Cells

Since acute stimulation with insulin inhibited IFNγ production in CD4+ cells, we asked if chronic hyperinsulinemia mirrored this effect. Hence, we compared the transcriptional profile of CD4+ cells of 12 non-diabetic RA patients with hyperinsulinemia and 44 patients having no hyperinsulinemia. This analysis revealed that hyperinsulinemia was frequently associated with downregulation of gene transcription in CD4+ cells. As much as 71% of all DEGs in CD4+ cells of patients with hyperinsulinemia were repressed (Appendix A). Notably, the dominating majority of DEGs in CD4+ cells of patients with hyperinsulinemia were also DEGs in CD4+ cells with high GI. The downregulated genes represented the biological processes of Regulation of Immune Processes (GO:0002682), Positive Regulation of T cell activation (GO:0050870), and Response to IFNγ signaling (GO:0034341). Indeed, the IFNG gene was the top gene repressed in CD4+ cells of patients with hyperinsulinemia (Figure 2J). It was accompanied by repression of transcription factors STAT4, TBX21, RORC, PRDM1, and EOMES essential for the Th1 phenotype and receptors assisting induction of IFNγ production, including integrins, Fcγ receptors, HLA-DRB1, and chemokine receptors (Figure 2J). Together, this transcriptional profile supported our experimental findings and provided a molecular basis for the arrest of IFNγ production, and suggested the reduced frequency of Th1 cells in patients with hyperinsulinemia.

In similarity with CD4+ cells with high GI, the cell cycle genes (Figure 3A) were deregulated in hyperinsulinemia (GO:0007049, FDR 2.39 × 10^−6^). This included upregulation of cyclin-dependent kinases CDK2, CDK4, and their inhibitor CDKN2D controlling G1/S transition (Figure 3B), which was experimentally affected by insulin in CD4+ cell cultures (Figure 2C) and a repression of CDK1 essential for G2/M transition of the cell cycle. Additionally, hyperinsulinemia inhibited the transcriptome required for cell cohesion MELK, CDCA7, ESCO2, mitotic spindle assembly KIF15, SGO1, and AURKB, and kinetochore formation KNL and NDC80 (Figure 3B). Transcription of all the genes correlated strongly with the GI (Spearman, rho 0.854–0.937, all *p* < 1.0 × 10^−25^), which rendered the cells susceptible to cell cycle arrest and senescence, as shown in the in vitro experiments.

### 3.5. Treatment with JAKi Promotes Insulin Signaling and Improves Glycolytic Index in CD4+ Cells of RA Patients

Among the analyzed transcriptome sets of CD4+ cells, 24 belonged to RA patients treated with JAK-inhibitors (JAKi), which inflicted anti-inflammatory effects by abrogating signaling from numerous cytokine receptors [29]. Thus, we investigated if JAKi interfered with the insulin effects on CD4+ cells. Comparing CD4+ cells of JAKi-treated and non-JAKi-treated patients (Appendix A), we found a significant upregulation of the insulin signaling genes including IRS1, IRS2, and AKT1 (Figure 3D), denoting an improvement in insulin sensitivity. Consequently, CD4+ cells of JAKi-treated patients had a higher glycolytic index despite the comparable insulin levels in those patients (Figure 3C,D, Appendix A). JAKi-treated patients presented upregulation of senescence controlling CDK inhibitors CDKN1A/p21, CDKN1C/kip2, CDKN2A/p16, and CDKN2D/p19, while the cell cycle promoting CDK1 and mitotic proteins remained repressed (Figure 3B).

### 3.6. JAK/STAT Signal Inhibition Promotes Insulin Effect and Induces Senescence in CD4+ Cell Cultures

Drawing a connection from the findings in JAKi-treated patients, we explored the immediate effect of the JAK/STAT inhibition in CD4+ cell cultures. Tracking the DNA content in JAKi-treated CD4+ cells, we found an accumulation of 7AAD+ DNA content in SSCs compared to the mock-treated cultures (Figure 3E,F), which occurred both in G1 and G2 phases (Figure 3E). Notably, we found that co-stimulation of CD4+ cells with JAKi and insulin significantly enhanced the accumulation of DNA content (Figure 3F,G). Additionally, JAKi significantly suppressed the CTV dilution, which disclosed a lower proliferation rate (Figure 3E), concordant with the low expression of cell division machinery genes in JAKi-treated patients (Figure 3B).

Analysis of CD4+ cells cultured with JAKi demonstrated an increase in IRS1 and IRS2 transcripts, which reproduced the improved insulin sensitivity observed in JAKi-treated patients (Figure 3H). Transcription of CDKN1A, CDKN2A, and CDK2D genes was significantly increased (Figure 3H). The increase in the DNA content together with the reduced proliferation rate, and activation of CDKN expression, characterized the senescent phenotype of the JAKi-treated cells and fully supported the CD4+ cell transcriptome of JAKi-treated patients. These experimental findings warranted a direct engagement of JAK/STAT signaling in the insulin sensitivity control. Importantly, the total cell number in the JAKi-treated CD4+ cell cultures was reduced (Figure 3G) and the anti-proliferative effect of JAKi remained unchanged after addition of insulin, which implied a gradual elimination of the senescent cells.

### 3.7. Effect of Hyperinsulinemia During JAK/STAT Inhibition in RA

To explore the additive effect of insulin and JAKi in RA patients, we compared CD4+ cells within the JAKi-treated (seven with hyperinsulinemia) and non-JAKi-treated groups (five with hyperinsulinemia). We found a significant suppressive effect of hyperinsulinemia on the glycolytic index in both groups (Figure 3D, Appendix A). Accordingly, hyperinsulinemia maintained its immunosuppressive effect in CD4+ cells of the JAKi-treated patients by mitigating upregulation of the key Th1 transcription factors RORC and PRDM1, as well as the chemokine receptors CXCR3 and CCR5 (Figure 4A). In contrast, we observed the additive effect of hyperinsulinemia and JAKi on the cell cycle-controlling genes CDKN2D/p19, EIF2S3B, GREM2, and YBX3 (Figure 4A). Consistent with the proposed elimination of senescent cells, we found no increase in serum IL6, IL8, or VEGF in JAKi-treated patients with hyperinsulinemia, while such an increase was significant in hyperinsulinemia of non-JAKi-treated patients (Figure 4B). Additionally, the serum level of survivin, taken as a proxy for cytolysis, was increased in the hyperinsulinemia patients, reaching significance in those treated with JAKi. Overall, the profile of CD4+ cells of JAKi-treated patients was characterized by a predominance of naïve T cell markers CCR7, IL2RG, and CD27, and the recent thymic emigrant T cell markers S1PR1, CR1, and PBXIP1 (Figure 4C), while expression of the memory T cell receptors CD44, CD69, IL2RA, and CD28 was low (Figure 4C). Thus, we concluded that JAKi and insulin had a traceable additive ability to induce senescence in Th1 cells, which facilitated replenishment of the recent thymic emigrants and naïve T cells to the blood of RA patients.

## 4. Discussion

In this study, we demonstrated that glycolysis in CD4+ T cells of RA patients is activated in direct relation to blood insulin levels and the insulin signaling machinery within these cells. This glycolytic activation facilitates glucose transport, cell cycle regulation, and cytokine production. Still, elevated glycolysis caused a proportional repression of IRS proteins, reducing insulin sensitivity in CD4+ cells.

The ability of insulin to regulate the cell cycle was validated in culture experiments. Insulin exposure in CD4+ cells promoted DNA content accumulation, recognition of DNA damage via γH2AX, and cell cycle arrest in the G1 phase, indicative of a senescent state. Senescence is tightly linked to aging and increased risk of T2D, metabolic syndrome, and cardiovascular disease [30]. In autoimmune disorders, T cell senescence impairs effector T cell differentiation and permits DNA damage accumulation, suggesting that senescent cells could be promising therapeutic targets in RA and systemic lupus erythematosus [31,32].

Our analysis of CD4+ T cells from RA patients with hyperinsulinemia revealed that insulin exerts a strong immunosuppressive effect on Th1 cells and regulates cell cycle progression, consistent with findings from in vitro experiments. In both cultured CD4+ cells and those isolated from patients with hyperinsulinemia, insulin inhibited IFNγ production by repressing key Th1 transcription factors and cell surface receptors essential for IFNγ signaling. These findings highlight an insulin-dependent mechanism contributing to T cell anergy and senescence. The IFN-dependent pathways are central to autoimmune diseases such as RA, SLE, T1D, and multiple sclerosis [33,34]. In RA, the IFN signature is associated with disease progression, skeletal damage, and response to antirheumatic drugs [35,36]. Our study showed a link between hyperinsulinemia and IFN pathway genes, emphasizing the biological context in shaping IFNγ effects. Insulin emerged as a significant contributor to these interactions.

Several studies have implicated IFN as a key driver of hyperinsulinemia and insulin resistance. For example, IFNα infusion has been shown to increase plasma insulin levels and delay insulin clearance, inducing insulin resistance in healthy individuals [37]. Similarly, IFNγ has been demonstrated to cause insulin resistance in muscle cells, promoting pancreatic insulin secretion during viral infections [38]. In diabetic mice, IFNγ deficiency improved insulin sensitivity [39].

Our results showed that blocking IFN signaling via JAK-STAT inhibition enhanced insulin sensitivity in CD4+ cells, boosting glycolysis and improving cell function. Enhanced insulin sensitivity was associated with increased transcriptional activity, improved cell cycle control, and better DNA damage repair. These findings suggest that promoting senescence in effector T cells could mitigate hyperinsulinemia in RA patients, protecting them from metabolic dysfunction.

Targeting the JAK-STAT pathway has shown benefits in regulating energy metabolism, reducing adiposity, improving insulin sensitivity, and mitigating inflammation. Evidence from clinical studies suggests that JAKi improves insulin sensitivity, as indicated by reduced glycated hemoglobin levels in RA patients with T2D [40] and non-diabetic individuals [41]. Our study provides experimental and clinical evidence that JAKi treatment enhances insulin signaling in CD4+ cells, promoting T cell anergy and senescence. Additionally, JAKi treatment was associated with the rejuvenation of the T cell population in RA patients, increasing the proportion of naïve CD4+ cells. In vitro, JAKi treatment synergized with hyperinsulinemia to suppress differentiation of CD4+ cells by inhibiting transcription factors essential for lineage maturation, thereby facilitating senescent effector cell elimination.

Interestingly, reduced insulin sensitivity in CD4+ cells may protect against the immunosuppressive effects of insulin, preserving effector T cell activity and enhancing autoimmunity. Population-wide studies in the USA, UK, and Taiwan have shown that antidiabetic drugs such as biguanides and dipeptidyl peptidase-4 inhibitors, which improve insulin sensitivity, are associated with a reduced risk of RA [42,43,44]. This suggests that repurposing antidiabetic drugs could be a viable therapeutic option in RA [45].

In summary, our study demonstrates that in RA, insulin exerts a potent immunosuppressive effect on effector Th1 cells and facilitates their elimination through senescence. These findings provide insights into potential therapeutic strategies targeting insulin signaling in autoimmune conditions.

## Figures and Tables

**Figure 1 cells-13-02124-f001:**
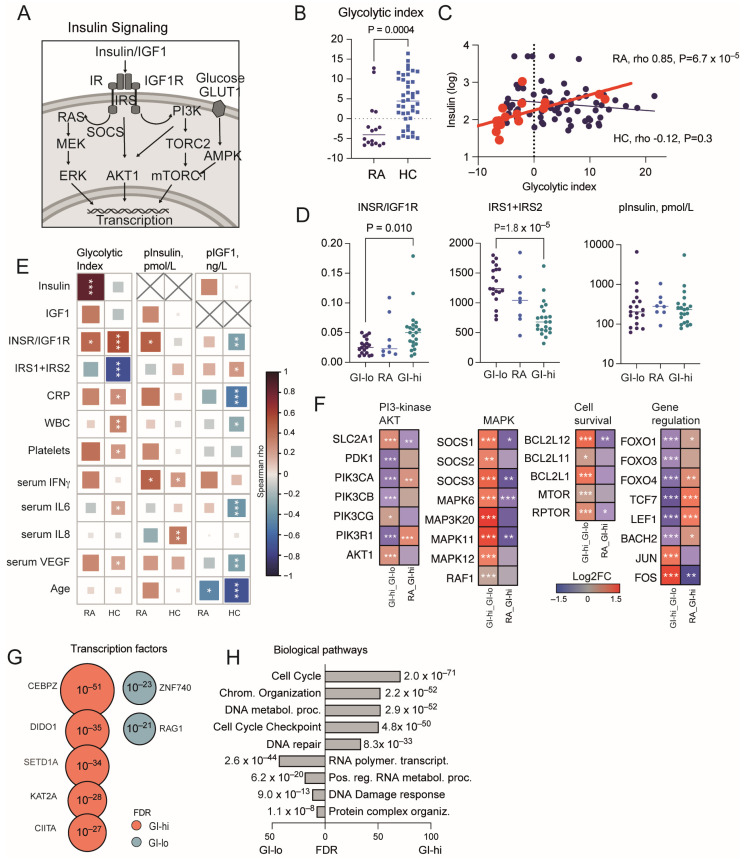
Glycolytic index of CD4+ T cells is in proportion to plasma insulin levels in rheumatoid arthritis. (**A**) Insulin signaling pathway. (**B**) Scatter plot of glycolytic index (GI) in CD4+ cells of untreated RA (n = 16) and age matched HC (n = 41). Solid line indicates median. Dotted line indicates average level. (**C**) Scatter plot of correlation between GI and plasma insulin levels of RA patients (red) and HC (black). Solid lines indicate lineal regression curve. (**D**). Scatter plot of gene expression in CD4+ cells. Solid line indicates median. (**E**) Heatmap of Spearman correlation rho values between GI, plasma insulin, and IGF1 levels with the insulin signaling genes and serological parameters. Color scale bar indicates rho value range. (**F**) Heatmap of the log2 fold change (FC) difference in gene expression of CD4+ cells with high (GI-hi, n = 34) and low (GI-lo, n = 35) GI, and between RA and HC with high GI, by RNA-Seq. *p*-values were obtained by DESeq2 test. Color scale bar indicates log2FC range. (**G**) Bubble diagram of transcription factor target enrichment (by FDR) among DEGs upregulated in CD4+ cells with high and low GI. (**H**) Bar diagram of biological processes (by GO:terms) enriched among DEGs upregulated in CD4+ cells with high and low GI. * *p* < 0.05, ** *p* < 0.01, *** *p* < 0.001. RA, rheumatoid arthritis; HC, healthy controls; CRP, C-reactive protein; WBC, white blood cell count; IGF1, insulin-like growth factor 1; INSR, insulin receptor; IGF1R, IGF1 receptor; IRS, insulin receptor substrate; IFNγ, interferon gamma; FDR, false discovery rate; GI, glycolytic index; VEGF, vascular endothelial growth factor.

**Figure 2 cells-13-02124-f002:**
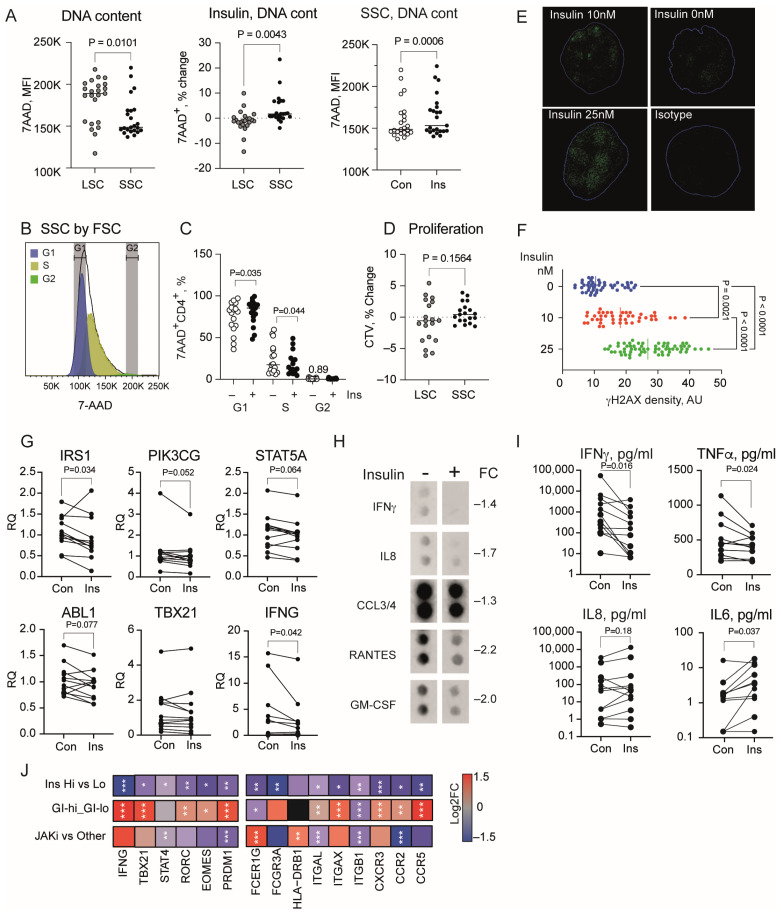
Insulin induces senescence and suppresses IFNγ production in CD4+ cells. CD4+ cells of 18 healthy subjects were stimulated with anti-CD3 and insulin 10 nM for 48 h. DNA content (7AAD) was analyzed by flow cytometry in large-size cell (LSC) and small-size cell (SSC) subsets. *p*-values were obtained by paired Wilcoxon test. (**A**) Scatter plot of DNA content by mean fluorescence intensity (MFI) of 7AAD+ cells. Scatter plot of DNA content change with insulin treatment. Con, control; Ins, insulin treated. Solid line indicates median. (**B**) Histogram of 7AAD+ cell distribution by phases of the cell cycle. Colored areas correspond to G1 (blue), S (yellow) and G2 (green) phases. (**C**) Scatter plot of 7AAD+ cell frequency in cell cycle phases. (**D**) Scatter plot of change in proliferation dye CellTrace violet (CTV) in insulin-stimulated CD4+ cells. Solid line indicates median. (**E**) Confocal microscopy image of nuclear γH2AX enrichment in insulin-treated THP1 cells. Blue line confines nuclear area. (**F**) Scatter plot of γH2AX density in nuclei of insulin-treated THP1 cells. Solid line indicates median. (**G**) Scatter plot of gene expression in insulin-treated CD4+ cells, in relative quantity (RQ) to control cell cultures. Expression was measured by qPCR. (**H**) Dotblot images of cytokine levels in pooled supernatants of insulin-treated and control cells measured by cytokine array. FC, fold change. (**I**) Scatter plot of cytokine protein levels in supernatants, by specific ELISA. (**J**) Heatmap of gene expression difference by log2 FC, by RNA-Seq. Samples are grouped by high (n = 34) and low (n = 35) glycolytic index (GI), high (n = 12) and low (n = 44) insulin, and JAKi-treated (n = 24) and non-JAKi-treated patients (Other, n = 32). *p*-values were obtained by DESeq2 test. Color scale bar indicates log2FC range. * *p* < 0.05, ** *p* < 0.01, *** *p* < 0.001.

**Figure 3 cells-13-02124-f003:**
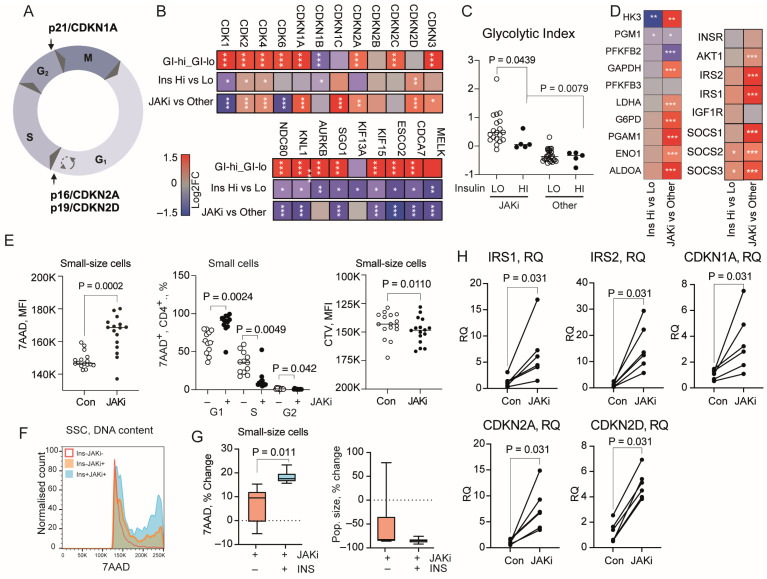
JAK/STAT-inhibitors (JAKi) increase insulin sensitivity and aggravate senescence in CD4+ cells. (**A**) Cell cycle phases and control check points. (**B**) Heatmap of transcription difference in CD4+ cell transcriptome of RA patients with high (n = 12) and low (n = 44) plasma insulin, JAKi-treated (n = 24) or non-JAKi-treated (Other, n = 32) and HC with high (n = 34) and low (n = 35) glycolytic index (GI), by RNA-Seq. Difference between groups was calculated by DESeq2-test. Color scale bar presents log2FC range. * *p* < 0.05, ** *p* < 0.01, *** *p* < 0.001. (**C**) Scatter plot of GI in CD4+ cells split by plasma insulin levels within JAKi-treated (Hi, n = 7, Lo, n = 17) and non-JAKi-treated (Other, Hi, n = 5, Lo, n = 27) RA patients. Solid line indicates median. (**D**) Heatmap of gene transcription difference in insulin signaling and GI of CD4+ cells as in B. (**E**) Scatter plot of DNA content by mean fluorescent intensity (MFI) in JAKi-treated and control cell cultures (n = 10). Scatter plot of frequency of 7AAD+ cells in different phases of the cell cycle. Scatter plot of proliferation tracer cell trace violet (CTV) intensity in JAKi-stimulated and control CD4+ cells. Solid line indicates median. (**F**) Histogram of 7AAD+ cell distribution in small-size CD4+ cells (SSC). Colored areas indicate cells stimulated with JAKi (yellow), JAKi + insulin (blue), and control culture (red line). (**G**) Box plot of DNA content change by 7AAD and population size change in paired CD4+ cell cultures stimulated with JAKi and insulin (n = 6). *p*-values by paired Wilcoxon test. Dotted line indicates basal level. (**H**) Scatter plot of gene transcription in CD4+ cells stimulated with anti-CD3 and tofacitinib (JAKi, 10 µM; Con, 0 µM) for 48 h. mRNA levels were analyzed by qPCR and are presented in relative quantity (RQ) to mock treated cells.

**Figure 4 cells-13-02124-f004:**
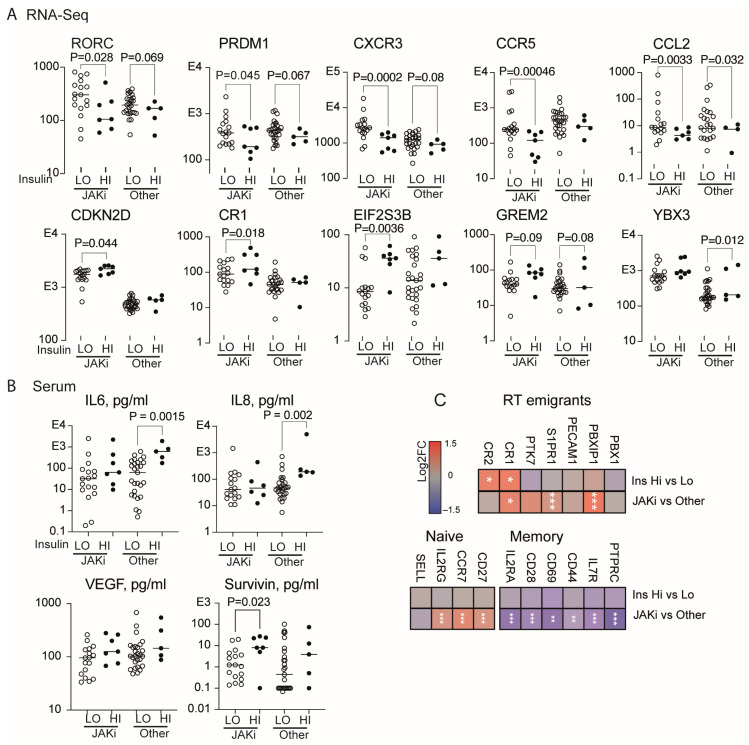
Insulin mitigates transcriptional effect of JAK-inhibitors (JAKi) in CD4+ T cells. CD4+ cells of RA patients were isolated by positive selection, activated on anti-CD3 for 2 h, and analyzed by RNA-Seq. Serum was used for protein measurements by immunosorbent assay. Differences between the groups were analyzed by DESeq2 test and Mann–Whitney U test. (**A**) Scatter plot of normalized gene transcription in JAKi-treated (Hi, n = 7, Lo, n = 17) and non-JAKi-treated patients (Other, Hi, n = 5, Lo, n = 27), split by high (Hi) and low (Lo) insulin levels. Solid line indicates median. (**B**) Scatter plot of IL6, IL8, survivin, and VEGF levels in serum. Solid line indicates median. (**C**) Heatmap of transcription difference by log2 fold change (FC) in CD4+ cells of RA patients with high (n = 12) and low (n = 44) plasma insulin levels, JAKi-treated (n = 24) or non-JAKi-treated (n = 32), by DESeq2. Color scale bar presents log2FC range. * *p* < 0.05, ** *p* < 0.01, *** *p* < 0.001. RT emigrants, recent thymic emigrants.

## Data Availability

Transcriptional data are available as Fastq-files and processed reads, deposited in Gene Expression Omnibus (GEO) at the National Centre for Biotechnology Information with the accession codes GSE201669 and GSE282517.

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
