# Peer review of "Insulin Sensitivity Controls Activity of Pathogenic CD4+ T Cells in Rheumatoid Arthritis"

_cells, 2024, doi:10.3390/cells13242124_

Round 1

Reviewer 1 Report

Comments and Suggestions for Authors

This study explored the impact of hyperinsulinemia on CD4+ T cell function in rheumatoid arthritis (RA) and its implications for autoimmunity. Insulin exposure increased the glycolytic index of CD4+ cells while suppressing insulin receptor substrate transcription, inversely correlating with insulin sensitivity—a relationship absent in healthy controls.

Insulin induced a senescent phenotype in CD4+ cells, characterized by cell cycle arrest, DNA enrichment, γH2AX accumulation, and a senescence-associated secretory phenotype (SASP), while also suppressing interferon-gamma (IFNγ) production. In hyperinsulinemic patients, this immune modulation was evident.

JAK-STAT signaling inhibition (via JAK inhibitors) restored insulin sensitivity, enhanced glycolysis, and promoted CD4+ cell senescence. In RA patients, JAK inhibitors increased naïve and thymic emigrant T cells in circulation, suggesting their role in mitigating RA pathology by facilitating pathogenic T cell clearance and leveraging insulin’s immunosuppressive effects. This finding highlights insulin’s dual role in metabolic and immune regulation and positions JAK inhibitors as potential therapeutic agents in RA.

The paper is very interesting and the Authors should be complimented for this timely research.

My only suggestion is to show individual data (Scatter plots) instead of box-and-whiskers graphs.

Author Response

This study explored the impact of hyperinsulinemia on CD4+ T cell function in rheumatoid arthritis (RA) and its implications for autoimmunity. Insulin exposure increased the glycolytic index of CD4+ cells while suppressing insulin receptor substrate transcription, inversely correlating with insulin sensitivity—a relationship absent in healthy controls.

Insulin induced a senescent phenotype in CD4+ cells, characterized by cell cycle arrest, DNA enrichment, γH2AX accumulation, and a senescence-associated secretory phenotype (SASP), while also suppressing interferon-gamma (IFNγ) production. In hyperinsulinemic patients, this immune modulation was evident.

JAK-STAT signaling inhibition (via JAK inhibitors) restored insulin sensitivity, enhanced glycolysis, and promoted CD4+ cell senescence. In RA patients, JAK inhibitors increased naïve and thymic emigrant T cells in circulation, suggesting their role in mitigating RA pathology by facilitating pathogenic T cell clearance and leveraging insulin’s immunosuppressive effects. This finding highlights insulin’s dual role in metabolic and immune regulation and positions JAK inhibitors as potential therapeutic agents in RA.

The paper is very interesting and the Authors should be complimented for this timely research.

My only suggestion is to show individual data (Scatter plots) instead of box-and-whiskers graphs.

Figures are now updated with scatter plots showing individual data within each figure.

Reviewer 2 Report

Comments and Suggestions for Authors

Congratulations! Your study is quite interesting and practical. Your findings about insulin signaling in RA could have a very beneficial potential for therapeutic applications in autoimmune conditions. Additionally, the findings about the metabolic ways of insulin in health open an important avenue for basic research. 

I liked the descriptions and figures very much. I would like to see shorter paragraphs, for instance, starting in line 268. Perhaps it would be better to have shorter paragraphs for each part of Fig 1. Also, the paragraph in lines 435-454 could be separated into two paragraphs. 

Author Response

Congratulations! Your study is quite interesting and practical. Your findings about insulin signaling in RA could have a very beneficial potential for therapeutic applications in autoimmune conditions. Additionally, the findings about the metabolic ways of insulin in health open an important avenue for basic research. 

I liked the descriptions and figures very much. I would like to see shorter paragraphs, for instance, starting in line 268. Perhaps it would be better to have shorter paragraphs for each part of Fig 1. Also, the paragraph in lines 435-454 could be separated into two paragraphs. 

Response: "Thank you for your positive input. Herewith, we improved the manuscript by introducing ne paragraphs in the M&M (2.2, line 110) and in Results (3:2, line 268; 3:5, line 402; 3:6, line 416)."        

Reviewer 3 Report

Comments and Suggestions for Authors

This is a good study on an important topic. A few suggestions to the authors:

-The authors must include a flow chart of study design outlining the clinical and experimental parts. This is very important as it will make it easier for readers to follow through with the narrative.

-Please mention what anticoagulant was used for plasma collection. It is also unclear why both serum and plasma were required.

-Please add results of ANOVA or Kruskal-Wallas to S1A.

-Please clarify in the methods the sampling strategy. Were subjects randomly selected? Stratified? Convenience sampling? Quota sampling? Include in the Discussion the limitations of the sampling strategy followed.

-Given the variability in patients’ medication intake, did the authors make any efforts to control for this variable? What about age, gender, and BMI?

-For insulin measurement, did the authors assay duplicates for each patient? This is important to account for technical variation.

-Why did the authors grow CD4+ T cells in 5% instead of 10% FBS?

-THP1 are of the myelogenous lineage unlike T cells. Why did the authors use it to assess γH2AX instead of the isolated T cells which were used for all other experiments? I think the authors need to detect γH2AX in CD4+ T cells to make the results coherent.

-Please mention what normality tests were used that made the authors decide to use non-parametric tests.

-Excellent figures.

Author Response

This is a good study on an important topic. A few suggestions to the authors:

-The authors must include a flow chart of study design outlining the clinical and experimental parts. This is very important as it will make it easier for readers to follow through with the narrative.

The study is based on two different, non-overlapping patient materials. One material contains untreated newly diagnosed RA patients and controls, the second material contains established RA patients. The experimental part of the study was done on healthy individuals. This is now clarified in the materials and methods section of the manuscript, lines 87-93. After these changes in the material description we believe that a flow chart is not needed.

-Please mention what anticoagulant was used for plasma collection. It is also unclear why both serum and plasma were required.

EDTA plasma was used for insulin measurements as required by the ELISA kit manufacturer. The information is added to the methods section of the manuscript.

-Please add results of ANOVA or Kruskal-Wallas to S1A.

We have now added unpaired t-test p-values for the comparison between HC and RA, and JAKi and non-JAKi treated patients.

-Please clarify in the methods the sampling strategy. Were subjects randomly selected? Stratified? Convenience sampling? Quota sampling? Include in the Discussion the limitations of the sampling strategy followed.

The material of HC and RA patients was randomly selected among the first visit patients with musculoskeletal complaints at the Rheumatology clinic, Sahlgrenska University Hospital. Patients of the established RA cohort were quota sampled by treatment with JAK-inhibitors.

-Given the variability in patients’ medication intake, did the authors make any efforts to control for this variable? What about age, gender, and BMI?

JAK inhibitor treated patients were quota sampled by number, gender and BMI. Unfortunately, the JAK inhibitor treated patients were not matched by age. See supplementary table S1A.

-For insulin measurement, did the authors assay duplicates for each patient? This is important to account for technical variation.

We agree, technical variation is an important issue. To guarantee technical stability in insulin measurements, the standard curve was run in triplicates on each plate. Unfortunately, we do not have insulin level measurements in the second sample of RA patients to confirm persistency of hyperinsulinemia.

-Why did the authors grow CD4+ T cells in 5% instead of 10% FBS?

This is routine cell culture media supplementation in our lab.

-THP1 are of the myelogenous lineage unlike T cells. Why did the authors use it to assess γH2AX instead of the isolated T cells which were used for all other experiments? I think the authors need to detect γH2AX in CD4+ T cells to make the results coherent.

We performed all these analyses in primary CD4+ T cells. The fidelity of γH2AX foci in nuclei of primary CD4 cells was insufficient for digital quantification by ImageJ software.

-Please mention what normality tests were used that made the authors decide to use non-parametric tests.

Many of the variables measured are indeed skewed with high outliers as demonstrated by deviating max and min values from median within the group. Additionally, the material is sometimes limited to 6-25 samples, we prefer using non-parametric analysis when possible, rather than normalizing the data by i.e. log-transformation.

-Excellent figures.

Thank you =)

Round 2

Reviewer 3 Report

Comments and Suggestions for Authors

The authors have adequately addressed my comments.